# Optimization of Bilayer Resistive Random Access Memory Based on Ti/HfO_2_/ZrO_2_/Pt

**DOI:** 10.3390/ma17081852

**Published:** 2024-04-17

**Authors:** Zhendong Sun, Pengfei Wang, Xuemei Li, Lijia Chen, Ying Yang, Chunxia Wang

**Affiliations:** 1College of Physics and Electronic Engineering, Chongqing Normal University, Chongqing 401331, China; mgzdsun01@163.com (Z.S.); 18716218339@163.com (X.L.); ljchen01@cqnu.edu.cn (L.C.); 20131184@cqnu.edu.cn (Y.Y.); 2Analog Foundries Co., Ltd., Chongqing 401332, China; pndn7712@163.com

**Keywords:** COMSOL, resistance switching, electrothermal coupling model, conductive filament

## Abstract

In this paper, the electrothermal coupling model of metal oxide resistive random access memory (RRAM) is analyzed by using a 2D axisymmetrical structure in COMSOL Multiphysics simulation software. The RRAM structure is a Ti/HfO_2_/ZrO_2_/Pt bilayer structure, and the SET and RESET processes of Ti/HfO_2_/ZrO_2_/Pt are verified and analyzed. It is found that the width and thickness of CF1 (the conductive filament of the HfO_2_ layer), CF2 (the conductive filament of the ZrO_2_ layer), and resistive dielectric layers affect the electrical performance of the device. Under the condition of the width ratio of conductive filament to transition layer (6:14) and the thickness ratio of HfO_2_ to ZrO_2_ (7.5:7.5), Ti/HfO_2_/ZrO_2_/Pt has stable high and low resistance states. On this basis, the comparison of three commonly used RRAM metal top electrode materials (Ti, Pt, and Al) shows that the resistance switching ratio of the Ti electrode is the highest at about 11.67. Finally, combining the optimal conductive filament size and the optimal top electrode material, the I-V hysteresis loop was obtained, and the switching ratio R_off_/R_on_ = 10.46 was calculated. Therefore, in this paper, a perfect RRAM model is established, the resistance mechanism is explained and analyzed, and the optimal geometrical size and electrode material for the hysteresis characteristics of the Ti/HfO_2_/ZrO_2_/Pt structure are found.

## 1. Introduction

Resistive random access memory (RRAM) is a new type of memory device, which uses the characteristics of transition metal oxide (TMO) film material of a resistive layer to change its internal electronic structure under the action of electric field and converts back and forth between high resistance and low resistance to store information [1,2]. Many research groups have extensively studied ZnO, HfO_2_, ZrO_2_, TiO_2_, CeO_2,_ Ta_2_O_5_, Nb_2_O_5_, and other TMOs for RRAM applications [3,4,5,6,7,8,9,10,11,12,13,14]. Among them, ZrO_2_ and HfO_2_ are considered to be the two most important materials due to their compatibility with complementary metal oxide semiconductors, high specific heat capacity, thermal conductivity, and dielectric constant [15]. Arun et al. [16] systematically studied the characteristics of resistance switches based on monolayer HfO_2_ using four different metal base electrode materials: Au, Al, Pt, and Cu. However, it was found that the resistance value of monolayer structure is not controllable, and there is no multi-value regulation effect. Moreover, Lee et al. [17] reported that compared with HfO_x_ monolayer devices, the ZrO_x_/HfO_x_ bilayer has a lower reset current, tight distribution of switching parameters, good switching durability up to 10^5^ cycles, and good data retention at 85 °C. After this report, Ismail et al. [18] researched an RRAM with a TaN/HfO_2_/ZrO_2_/Pt structure, where a bilayer HfO_2_/ZrO_2_ thin film structure was deposited by radio frequency sputtering at room temperature (RT) to investigate the resistive switching characteristics and the switching ratio > 10 for 10^3^ DC cycles was calculated. Therefore, in this paper, HfO_2_ and ZrO_2_ are used as the transition layers of the bilayer RRAM structure to study the resistance mechanism behind it.

In order to study the resistance mechanism of RRAM devices, many previous RRAM models with different characteristics and accuracies have been proposed [19,20,21,22,23,24,25]. Menenzel et al. [25] focused on the effect of the electric field on ion migration, considering the thermal properties of the upper and lower electrode materials: Ti/Nb:STO (Nb-doped SrTiO_3_) of RRAM devices. They estimated the temperature characteristics of conductive filaments based on experimental data and established an electrothermal 2D finite element model. In the model, the effect of temperature and electric field on the change in SET time with the change in voltage is obtained by simulating the change in the resistivity in the filament region, but the bipolar I-V conversion characteristics of the device were not obtained. Kim et al. [26] modeled and simulated a bilayer Pt/Ta_2_O_5_/TaO_x_/W RRAM device. The model based on temperature and electric field accelerated oxygen vacancy migration, considering the effect of oxygen vacancy generation rate on the resistance characteristics of the device, obtained the DC bipolar I-V characteristics of the device. However, the recombination of oxygen vacancies and oxygen ions is neglected. Moreover, the transient properties of the Joule thermal model are ignored, and the temperature of the resistive lattice does not change with time. Huang et al. [27] proposed a physical analysis model of metal oxide RRAM under both DC and pulse operating modes. The model covers the transport of oxygen vacancies and oxygen ions, the temperature distribution under heat conduction, and also studies the influence of the width of conductive filaments and the gap between conductive filaments and electrodes, but the thickness of the transition layer is not considered.

Therefore, based on the model of Huang et al., this paper refers to a bipolar RRAM numerical model based on temperature field accelerated ion migration proposed by Larentis et al. [23] and takes into account the generation and recombination of local oxygen vacancy in the medium layer, and then establishes a perfect Ti/HfO_2_/ZrO_2_/Pt electric–thermal coupling model. On the Ti/HfO_2_/ZrO_2_/Pt model, the scanning voltage rate is set to 100 mV/s by applying a cycle of forward and reverse scanning voltage. The distribution of oxygen vacancy concentration (n_D_), temperature (T), and electric potential (Ψ) in Ti/HfO_2_/ZrO_2_/Pt RRAM with an applied voltage device is discussed in detail. The two-dimensional distribution diagram and one-dimensional curve diagram are used to show the distribution details of the three physical quantities. Thus, the RESET and SET process and mechanism of RRAM under the Ti/HfO_2_/ZrO_2_/Pt double structure are verified. In this paper, not only are the width of the bilayer conductive filament studied but also the thickness of the transition layer and the influence of the commonly used metal top electrode materials (Ti, Pt, and Al) on the hysteresis characteristics of the device. By comparison, it is found that under the conditions of the width ratio of the conductive filament to the transition layer (6:14) and the thickness ratio of HfO_2_ to ZrO_2_ (7.5:7.5), Ti is selected as the top electrode material. In this case, the RRAM has a stable high-low resistance state, that is, the maximum switching ratio and obvious hysteresis characteristics. Thus, the optimization of RRAM under the Ti/HfO_2_/ZrO_2_/Pt structure is completed by changing the electrode structure of RRAM and the size between the switching layer and the conductive filament in the simulation work.

## 2. Simulation Methods

In this paper, the three differential equations of oxygen vacancy concentration (n_D_), temperature (T), and electric potential (Ψ) are simulated by using custom coefficient-type partial differential equations. The following are the physical fields and partial differential equations involved in the simulation model of RRAM devices.

### 2.1. Electrical–Thermal Coupling Physical Model

In order to study the change in oxygen ion concentration during the change in resistance value, oxygen ion migration is represented by the migration equivalent of oxygen vacancy. The movement of oxygen vacancies mainly includes three aspects: the diffusion movement of oxygen vacancies, the drift movement under the action of voltage, and the formation and recombination of local oxygen vacancies in the dielectric layer. The migration of oxygen vacancy is expressed by the flow rate of oxygen vacancy per unit volume, including drift and diffusion, so the diffusion and drift flux can be used to represent the migration flux of oxygen vacancy. When the voltage is 0 V, the drift motion does not need to be considered, and the oxygen vacancy will diffuse from the place with high concentration to the place with low concentration, and the diffusion motion will dominate at this time. Once the voltage is applied, the atomic transition barrier will decrease so that the oxygen vacancy can obtain enough energy to jump out of the original position. The reduction in the potential barrier is related to the applied potential, and the proportional relationship is qbE, where q is the amount of charge, b is the minimum size of the partition lattice, which is also the size of the grid in the simulation, and E is the electric field strength. The atomic transition probability is exponentially related to the barrier level, and the oxygen vacancy will jump rapidly after the voltage is applied, thus increasing the drift current, and the drift motion is dominant at this time. Based on the above analysis, the expression of oxygen vacancy migration is as follows [23]:(1)$$j=j_{diff}+j_{dift}=-D\nabla n_{D}+\nu n_{D}$$where *j_diff_* and *j_dift_* represent diffusion and drift flux, respectively, the unit is cm^2^s^−1^, *n_D_* is the oxygen vacancy concentration, the unit is cm^−3^, *D* is the oxygen vacancy diffusion coefficient, *v* is the oxygen vacancy drift rate, and the unit is cm^2^s^−1^. Therefore, the rate of change in oxygen vacancy per unit volume with time, namely the drift–diffusion equation of oxygen vacancy, can be expressed as follows [23]:(2)$$\frac{\partial n_{D}}{\partial t}=-\nabla \cdot j+G-R=\nabla (D\nabla n_{D}-\nu n_{D})+G-R$$
and because the diffusion coefficient of ions is related to temperature, it satisfies the Arrhenius equation [23]:(3)$$D=D_{0}\cdot \exp(-E_{a}/kT)$$
(4)$$D_{0}=\frac{1}{2}\cdot a\cdot f^{2}$$

*D_0_* is the exponential factor of the diffusion coefficient, *E_a_* is the oxygen vacancy diffusion activation energy, *k* is the Boltzmann constant, *T* is the temperature, a is the ion transition distance in nm, and *f* is the escape frequency.

The drift rate of oxygen vacancy also obeys the Temperature-dependent Arrhenius equation [28]:(5)$$\nu =2af\exp(-\frac{E_{a}}{kT})\sinh(\frac{qbE}{2kT})$$

The reason for the generation and recombination of local oxygen vacancies is that the concentration of oxygen vacancies in the medium layer is uneven, resulting in excessive electric field intensity at the local position after the voltage is applied, which breaks the covalent bond of HfO_2_ molecules and forms HfO_x_ and free oxygen ions [24]. Oxygen ions can combine with each other to form oxygen molecules and free to the trap at the interface between the oxide and the metal, and HfO_x_ will introduce new oxygen vacancies so that the concentration of oxygen vacancies in the local area becomes higher, that is, the effect. On the contrary, the recombination of oxygen vacancies will occur [29]. The expression of oxygen vacancy generation and recombination is as follows [27]:(6)$$G=A\cdot \exp((-E_{b}-qbE)/kT)$$
(7)$$R=A\cdot \exp(-E_{c}/kT)$$
where *A* is the constant related to vibration frequency, *E_b_* is the average activation energy of oxygen vacancy, and *E_c_* is the relaxation energy in the recombination process.

Finally, as mentioned above, Equations (3)–(7) are brought into the drift–diffusion Equation (2) for calculation, and the values of the parameters involved in COMSOL are shown in Table 1.

#### 2.1.1. Electrical Conduction Model

Current continuity equation [31]:(8)∇⋅σ∇ψ=0
where *σ* is the conductivity of the conducting filament, expressed in Ω^−1^ cm^−1^, and *Ψ* is the potential. In order to solve (2)–(8), we need to give a model of the conductivity *σ*, which can be expressed as follows:(9)$$\sigma =\sigma_{0}\exp(-E_{AC}/kT)$$
where *σ*_0_ is the pre-exponential factor, the unit is Ω^−1^ cm^−1^; *E_AC_* is the conductive activation energy. Suppose that *σ_0_* and *E_AC_* both depend on the oxygen vacancy concentration. *σ*_0_ is the equation for oxygen vacancy concentration; the specific setting is explained in the following text.

#### 2.1.2. Joule Heat Model

The heat conduction equation is as follows [31]:(10)$$\rho C_{P}\frac{\partial T}{\partial t}-\nabla \cdot k_{th}\nabla T=Q=J\cdot E=\sigma {\left|\nabla \psi \right|}^{2}$$where *ρ* is the density, *C_p_* 
is the heat capacity, *k_th_* is the thermal conductivity of 
conductive filament, and its unit is Wm^−1^K^−1^. It is also 
assumed that the change in *k_th_* depends on the oxygen vacancy 
concentration. The steady-state mode is exclusively considered, disregarding $\rho C_{P}\frac{\partial T}{\partial t}$. By applying 
Fourier’s law, the heat flux formula can be derived as follows: $J_{T}-k_{th}\cdot \nabla T$. The left side of the equation is to find its divergence. Because there is a passive field in the medium layer, the steady-state heat conduction equation is obtained as follows:(11)$$\nabla \cdot (-k_{th}\nabla T)=0$$

The COMSOL simulation calculation process is the solution process of the partial differential Equations (2), (8) and (11), so as to calculate the distribution of oxygen vacancy concentration *n_D_*, electric potential *Ψ*, and temperature *T* during the device conversion process.

### 2.2. Model Establishment in COMSOL

#### Geometric Model

In this paper, the 2D axisymmetric structure of COMSOL is used to create a geometric model of RRAM, as shown in Figure 1, where the left boundary (r = 0) is the axis of rotational symmetry. The device width, thickness, and top electrode to adjust material size parameters are shown in Table 2.

In the model, the same width and thickness of the top and bottom electrodes were set according to the nanoscale dimensions in ref. [32] where the width is twice greater than described in the literature, and the thickness is half that. The width and thickness of CF1 and HfO_2_, CF2, and ZrO_2_ were initially set, but these parameters will be changed in our simulation work. The sum of the thickness of the Y_2_O_3_ switching layer and Y metal layer in ref. [32] is taken as the thickness of the HfO_2_ layer and ZrO_2_.

### 2.3. Definition of Material Properties

According to the parameters required in the analysis of the electro-thermal coupling model, the material properties of each layer entity used in the simulation of Ti/HfO_2_/ZrO_2_/Pt RRAM devices are given in Table 3. In the simulation, it is assumed that the heat capacity C_p_, density ρ, and relative permittivity of CF1 and CF2 are equal to those of HfO_2_ and ZrO_2_, and their conductivity σ and thermal conductivity k_th_ are defined as a function of oxygen vacancy concentration n_D_.

According to the calculation formula of atomic density,
(12)$$P=\rho /(\frac{m_{mol}}{N_{A}})$$

The molar mass of HfO_2_ was 210.49 g/mol, ZrO_2_ was 123.218 g/mol, and N_A_ was Avogadro’s constant 6.02 × 10^23^ /mol. The atomic density P_1_ of HfO_2_ = 2.77 × 10^22^ cm^−3^ and P_2_ of ZrO_2_ = 2.88 × 10^22^ cm^−3^ can be obtained by Equation (12). Since the maximum doping concentration can reach 4.3% of the atomic density [23], the maximum oxygen vacancy concentration of HfO_2_ in the simulation is 1.2 × 10^21^ cm^−3^, and the maximum oxygen vacancy concentration of ZrO_2_ is 1 × 10^21^ cm^−3^. According to Equation (9), the conductivity σ can be determined by defining σ_0_ in relation to E_AC_ and oxygen vacancy concentration n_D_. The σ_0_ of HfO_2_ is estimated from 10 Ω^−1^ cm^−1^ to 3300 Ω^−1^ cm^−1^, respectively, by high and low resistance resistivity. The σ_0_ of ZrO_2_ varies from 10 Ω^−1^ cm^−1^ to 3000 Ω^−1^ cm^−1^. Figure 2a shows the relationship between σ0 and oxygen vacancy concentration. The E_AC_ of HfO_2_ activation energy in high and low resistance states are 0.087 eV and 0.018 eV, respectively. And the E_AC_ of ZrO_2_ is 0.087 eV and 0.018 eV, respectively. Figure 2b shows the relationship between E_AC_ and oxygen vacancy concentration. When the oxygen vacancy concentration reaches half of the maximum concentration, conductive filaments are formed, and the device is converted to a low resistance state [27]. In the simulation, the thermal conductivity *k_th_* at the filament changes from the thermal conductivity of HfO_2_ and ZrO_2_ to that of Hf and Zr with the increase in oxygen vacancy concentration. Because the oxygen vacancy concentration n_D_ is assumed to enhance thermal conductivity due to the free-carrier contribution to heat conduction. As shown in Figure 2c, based on the Wiedemann–Franz law, we assumed linear dependence of *k_th_* on n_D_ [22,23]. In addition, the above processes are considered to be linear changes.

## 3. Results and Discussion

RESET is a process where the device shifts from a low resistance state to a high resistance state. When the COMSOL simulation model is established, the initial conditions of the device are that the conductive filament is fully connected; the boundary temperature and initial temperature are room temperature 300 K; the top electrode has a scanning voltage of 0 V→1 V→0 V, where the scanning second rate is 100 mV/s; the bottom electrode is grounded. In order to reflect the resistance process, CF1 and CF2 regions are considered, and the distribution diagram of oxygen vacancy concentration, potential, temperature, and other parameters of the device is obtained under the forward bias voltage of 0.4 V–0.8 V (step by 0.1 V).

When the bias voltage is 0.4 V, the conductive wire morphology in Figure 3a shows no significant change, and CF1 is still in a state of conduction. When the voltage is increased to 0.5 V, the oxygen vacancy concentration in CF1 changes, but no fracture occurs at this time. It can be seen from Figure 3b that the lowest oxygen vacancy concentration in the CF1 layer is about 1.05 × 10^21^ cm^−3^ (>0.6 × 10^21^ cm^−3^). Due to the change in oxygen vacancy concentration, CF1 begins to gradually break down. The voltage continues to increase to 0.6 V, and it can be intuitively seen from Figure 3b that CF1 has a clear fracture zone (the region with a blue concentration of 0 in CF1). When CF1 continues to increase the voltage to 0.8 V after fracture, Figure 3d,e show that the fracture region further expands. We simulated the distribution of oxygen vacancy concentration in the conductive filament region of RRAM at a forward bias voltage of 0.4 V–0.8 V (step by 0.1 V) in the z direction, as shown in Appendix A. Combined with the one-dimensional distribution curve in Appendix A, it can be found that the conductive wire morphology clearly breaks at z = 28 nm, the maximum fracture width is about 2 nm, and the V_RESET_ voltage is 0.6 V.

In order to further study the temperature distribution, it can be seen from Figure 4a,b that when the applied voltage is 0.4 V and 0.5 V, the temperature distribution is uniform and mainly distributed within CF1. Figure 4c–e show that as the voltage increases, the temperature is no longer uniformly distributed. We further calculate the distribution of temperature and electric potential in the z direction when the forward bias is 0.4 V–0.8 V (step by 0.1 V), as shown in Appendix A. The specific distribution law can be clearly seen from the one-dimensional distribution curve in Appendix A: When the temperature increases caused by Joule heat, the one-dimensional distribution curve of temperature is different from the parabola in the initial stage, and it is no longer smooth and distorted. Specifically, the temperature near the ZrO_2_ interface is low, and the temperature near z = 28 nm is high. In the region where CF1 is broken, the temperature can reach up to 950 K. The results show that in this region, the migration rate of oxygen vacancy is fast, and CF1 will break at the highest temperature.

Then, the distribution of potential in the RESET process is studied. Figure 5 shows the distribution of potential in the dielectric layer under different forward bias voltages. As can be seen from Figure 5a,b, when the forward voltage is 0.4 V and 0.5 V and CF1 is not broken, the oxygen vacancy concentration in the medium layer is the same, so the potential distribution is uniform, and the maximum potential is at the top of the upper half of the region. Figure 5c,e show that after CF1 breaks, the inner potential of the upper half of the region increases, indicating that the resistance here increases and the partial voltage increases. Appendix A shows that the potential is uniformly distributed when CF1 is switched on. When the fracture is complete, the electric potential in the fracture area increases sharply, and the resistance value becomes large, showing a high resistance state.

In summary, in the RESET process, combined with the distribution of oxygen vacancy concentration, temperature, and potential, the whole process of CF1 from conduction to fracture can directly or indirectly indicate the fracture partitions the top electrode and the bottom electrode regions, forming a high resistance state. The device completes the RESET process from the low resistance state to the high resistance state.

We further verified that not only the RESET process but also the SET process exists in the Ti/HfO_2_/ZrO_2_/Pt structure. In this paper, the bias voltage is set to the reverse value, and the bias voltage of the top electrode is set to 0 V→−1 V→0 V, where the sweep second rate is still 100 mV/s, and the oxygen vacancy concentration distribution under the reverse bias voltage is obtained, as shown in Figure 6. Figure 6a,b show that the device is still in the high resistance state of the previous process, CF1 is in the broken state, and the oxygen vacancy concentration in the broken region is low. With the increase in voltage, the concentration of oxygen vacancy in the fracture gap of the filaments increases, and then CF1 recombines in the fracture zone and re-conducts. As shown in Figure 6c, the conductive wire breaks again when the voltage is increased to −0.7 V.

The temperature distribution in the SET process of RRAM is shown in Figure 7. In the initial stage, the temperature distribution during CF1 fracture is mainly concentrated at the fracture of the red high-temperature region in Figure 7a,b. The current at the fracture is small, so less Joule heat is generated. With the increase in voltage, the current through z = 28 nm increases, and the oxygen vacancy accumulates toward the fracture, so Joule heat is also generated. Then, the high-temperature area at CF1 moves toward the ZrO_2_ layer, and the temperature is gradually evenly distributed throughout the device. As shown in Appendix A, after calculating the 2D distribution of oxygen vacancy concentration, we simulated the distribution of temperature and potential in the z direction when applying the reverse bias voltage of −0.85 V, −0.8 V, −0.7 V, −0.6 V, and −0.5 V. Appendix A is the one-dimensional temperature distribution curve of the device during the SET process. Combined with Appendix A, it can be determined that the specific location of the fracture region is in CF1, where the temperature is low near z = 28 nm in the z direction, while in CF1, the temperature is high at z = 24 nm, and the migration rate of oxygen vacancy is accelerated.

After the above discussion of the temperature distribution of RRAM devices, further attention should be paid to the distribution of potential. Figure 8a,b and Appendix A are two dimensional and one-dimensional distribution curves of potential distribution. Figure 8a,b show that the electric potential is mainly concentrated at the red high temperature near the top electrode Ti, showing a high resistance state. Thereafter, Figure 8a–c show that with the increase in voltage, the potential in the lower half region of CF1 is no longer evenly distributed but increases in the inner potential, indicating that the resistance value of the resistance in the lower half region becomes larger. Appendix A can directly reflect the position change in the CF1 fault region: the abrupt increase in the fault region changes from the previous steep increase region at z = 28 nm to z = 24 nm. The uniform distribution of potential in the previous fracture region indicates that CF1 has a process from fracture to recombination fracture.

In summary, the distribution of oxygen vacancy concentration, temperature, and potential under reverse bias voltage can all indicate that CF1 breaks during RESET to re-conduction during SET, and the device converts from the above high resistance state to a low resistance state, completing the SET process.

After verifying the existence of REST and SET resistive processes, this paper further studies the effect of conductive filament size on RRAM. CF1 and CF2 in the 2D geometric model of the device are set to different widths and thicknesses; that is, the radius of the conductive filaments is set to 5 nm, 6 nm, 7 nm, and 8 nm, and the thickness takes the thickness of the HfO_2_ layer as an example (the total thickness of HfO_2_ layer and ZrO_2_ layer is 15 nm) and is set to 6.5 nm, 7 nm, 7.5 nm, and 8 nm, respectively, for forward and reverse scanning of the device. The results are shown in Figure 9a,b. The radius of CF1 and CF2 reflects the limiting current of the SET process and the resistance of the low resistance state. Figure 9a shows that with the increase in the radius of CF1 and CF2, the RESET conversion current increases, and the limiting current and low-resistance resistance of the SET process also increase. Therefore, it can be concluded that the larger the RESET conversion current, the larger the SET limiting current. However, in Figure 9a, the hysteresis curves under the reverse bias voltage almost coincide at 7 nm and 8 nm, which also indicates that the influence of excessive conversion current is the instability of the high–low resistance state and the reduction in the switching ratio. Figure 9b shows that with the increase in HfO_2_ thickness, the RESET conversion current gradually decreases, but the hysteretic characteristics have no specific trend, and the most obvious high–low resistance state and the maximum switching ratio exist at 7.5 nm. Therefore, 7.5 nm is considered to be the best thickness for HfO_2_ and ZrO_2_ and has the most stable high and low resistance state at this thickness.

In the model of RRAM devices, in addition to the influence of the size of CF1 and CF2 on its performance, the material selection of the top electrode also affects the I-V characteristics of the device. In this paper, the influence of commonly used metal top electrode materials on the resistance transition characteristics of the device is studied. As shown in Figure 10, the electrode combination needs to be changed into three structures of Pt-Pt, Ti-Pt, and Al-Pt in the analysis process to draw I-V curves for comparison. The results show that the peak current of the Ti/HfO_2_/ZrO_2_/Pt device is about 0.25 mA in the SET process and 0.14 mA in the RESET process with Ti-Pt as an electrode. The switching ratio R_off_/R_on_ of devices with electrode combinations of Pt-Pt, Ti-Pt, and Al-Pt are 1.97, 11.67, and 2.35, respectively. Ti-Pt has the largest switching ratio, so the Ti/HfO_2_/ZrO_2_/Pt structure has a more stable high–low resistance state in comparison.

Based on the above discussion of the size of CF1 and the top electrode material, a bilayer RRAM with a Ti/HfO_2_ (thickness 7.5 nm)/ZrO_2_ (thickness 7.5 nm)/Pt structure is adopted in this paper, where the radius width of CF1 is 6 nm. The hysteresis curve, as shown in Figure 11, can be obtained. The forward voltage of 0 V→1 V→0 V is applied to the device, and the V_RESET_ voltage of RRAM is 0.6 V from conduction to disconnection. When the voltage is increased to 1 V, the RESET process of the device ends, completing the transition from a low resistance state to a high resistance state and then maintaining a high resistance state. Then, the reverse voltage is increased from 0 V to −1 V, and when the voltage is increased, the transformation process from the high resistance state to the low resistance state begins until the CF1 fracture region moves down to the ZrO_2_ interface when the voltage increases to −0.7 V. The SET process from high resistance state to low resistance state is completed when the voltage is −0.7 V, that is, the V_SET_ is −0.7 V. Combined with the curve in Figure 11, it can be calculated that the average resistance value of the high resistance state (R_off_) of the device is 43.10 kΩ, the average resistance value of the low resistance (R_on_) state is 4.12 kΩ, and the switching ratio is R_off_/R_on_ = 10.46. Compared to the experimental data of Ismail et al.’s TaO_2_/ZrO_2_/HfO_2_ /Pt bilayer structure [18] in Table 4, the simulated switching ratio, SET voltage, and RESET voltage for a similar structure are shown in the table. By comparison, it can be observed that in spite of choosing a smaller top electrode size (15 nm in this study compared to Ismail et al.’s 70 nm), the switch ratio differs by approximately 3. Not only that but our model also has lower SET and RESET voltages.

## 4. Conclusions

In this paper, firstly, a bilayer RRAM model of Ti/HfO_2_/ZrO_2_/Pt was established using COMSOL Multiphysics, and the RESET and SET resistance processes of RRAM, as well as the distribution of oxygen vacancy concentration (inside CF1 and CF2), temperature, and potential of the two processes were verified. Then, the effects of the size of the conductive filament and the top electrode material on the performance of the device were investigated. Under the conditions of the width ratio of the conductive filament to the transition layer (6:14) and the thickness ratio of HfO_2_ to ZrO_2_ (7.5:7.5), the RRAM has a stable high–low resistance state. In addition, the results also show that the RRAM’s larger CF1 and CF2 size is not better. Moreover, a larger conversion current will appear at the same time that the switching ratio is lower, and the high and low resistance states are not obvious. Next, we set Pt-Pt, Ti-Pt, and Al-Pt as the electrodes and calculated that the switching ratios were 1.97, 11.67, and 2.35, respectively. So, the comparison showed that the switching ratio of the Ti electrode was the largest, which was 11.67. Finally, we found that the switching ratio R_off_/R_on_ of Ti/HfO_2_/ZrO_2_/Pt RRAM is 10.46 by taking the optimal size and selecting the best top electrode material. And comparing our simulation results with the previous three HfO_2_/ZrO_2_ bilayer RRAM [7,18,38], we can see that the smaller top electrode can also achieve an approximately equal switching ratio (about 3 less than Ismail et al.’s RRAM structure), which is more advantageous and valuable in low-power applications and integrated circuits.

## Figures and Tables

**Figure 1 materials-17-01852-f001:**
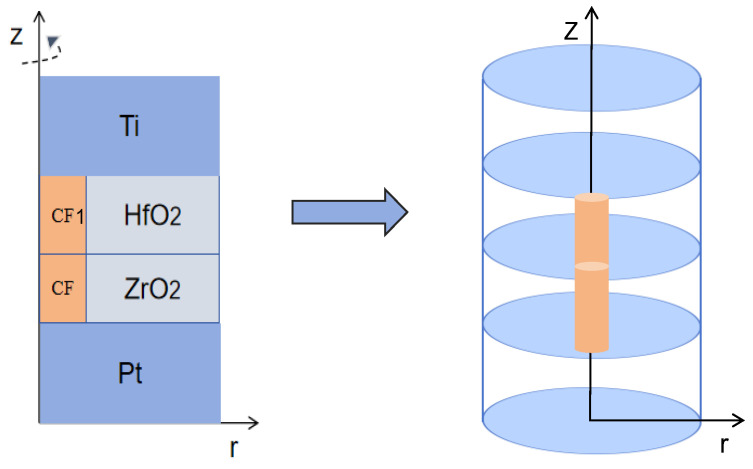
Schematic diagram of the 2D axisymmetric geometric model of the device. In COMSOL, the *Z*-axis is used as the rotation axis, and the 2D model is rotated 360° to obtain the solid model.

**Figure 2 materials-17-01852-f002:**
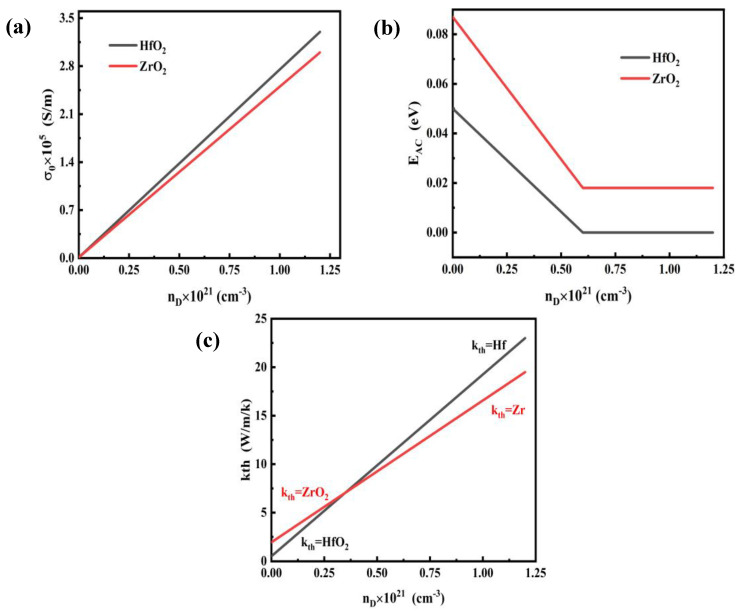
Customize material propertiesstomize material properties in COMSOL: (**a**) conductivity index pre−factor σ_0_; (**b**) conductivity activation energy E_AC_; (**c**) thermal conductivity k_th._

**Figure 3 materials-17-01852-f003:**
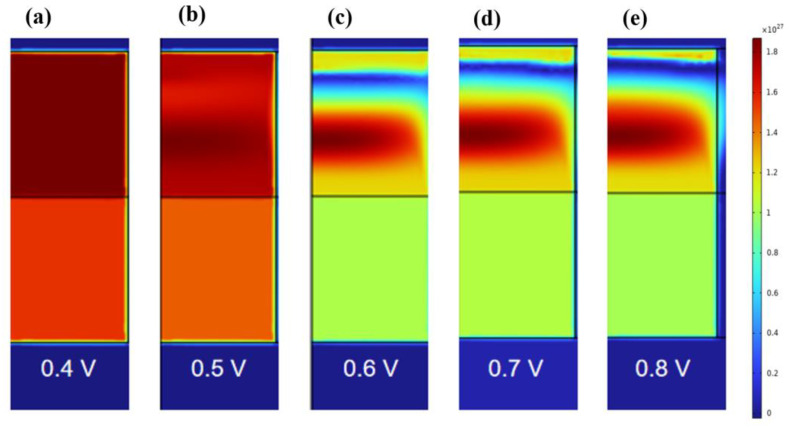
Two−dimensional distribution of oxygen vacancy concentration inside CF1 and CF2 of Ti/HfO_2_/ZrO_2_/Pt at the bias voltage of 0.4 V–0.8 V (step by 0.1 V) in the SET process.

**Figure 4 materials-17-01852-f004:**
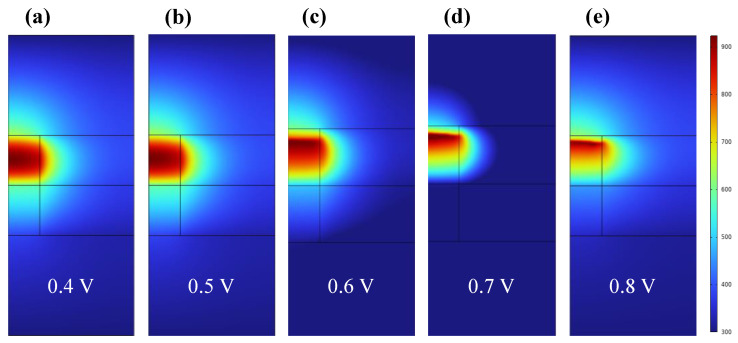
Two−dimensional temperature distribution in the SET process of Ti/HfO_2_/ZrO_2_/Pt at bias voltage 0.4 V–0.8 V (step by 0.1 V).

**Figure 5 materials-17-01852-f005:**
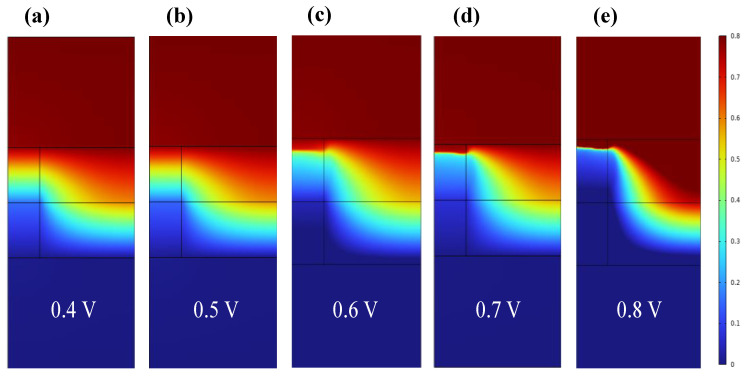
Two−dimensional distribution of potential in the SET process of Ti/HfO_2_/ZrO_2_/Pt at bias voltage 0.4 V–0.8 V (step by 0.1 V).

**Figure 6 materials-17-01852-f006:**
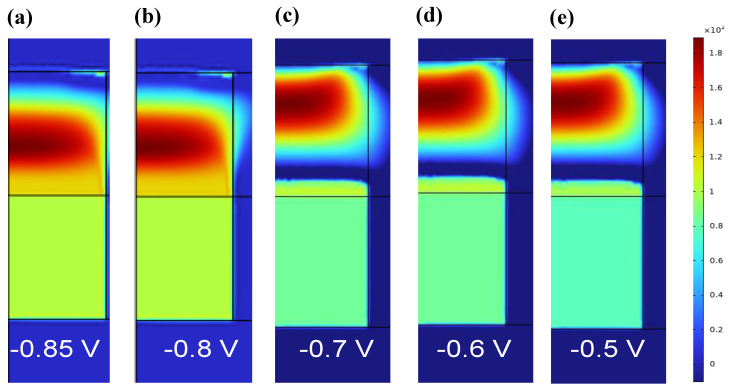
Two−dimensional distribution of oxygen vacancy concentration inside CF1 and CF2 at the bias voltage of −0.85 V, −0.8 V, −0.7 V, −0.6 V, and −0.5 V in the RESET process of Ti/HfO_2_/ZrO_2_/Pt.

**Figure 7 materials-17-01852-f007:**
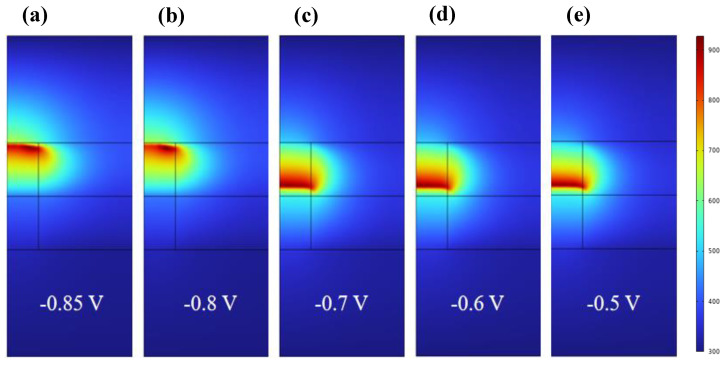
Two−dimensional temperature distribution in the RESET process of Ti/HfO_2_/ZrO_2_/Pt at bias voltages −0.85 V, −0.8 V, −0.7 V, −0.6 V, and −0.5 V.

**Figure 8 materials-17-01852-f008:**
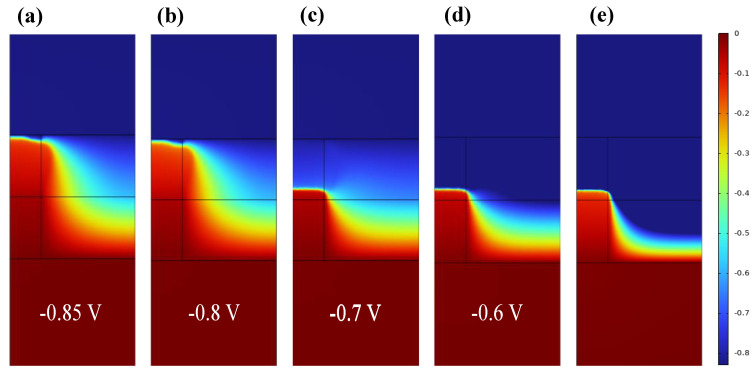
Two−dimensional distribution of potential in the RESET process of Ti/HfO_2_/ZrO_2_/Pt at bias voltages −0.85 V, −0.8 V, −0.7 V, −0.6 V, and −0.5 V.

**Figure 9 materials-17-01852-f009:**
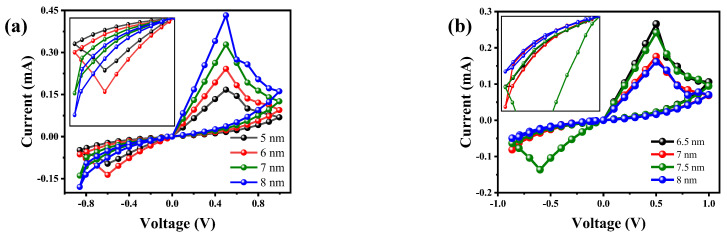
Influence of size on RRAM device performance: (**a**) the width of the CF1 layer at 5 nm, 6 nm, 7 nm, and 8 nm; (**b**) the thickness of the CF1 layer at 6.5 nm, 7 nm, 7.5 nm, and 8 nm.

**Figure 10 materials-17-01852-f010:**
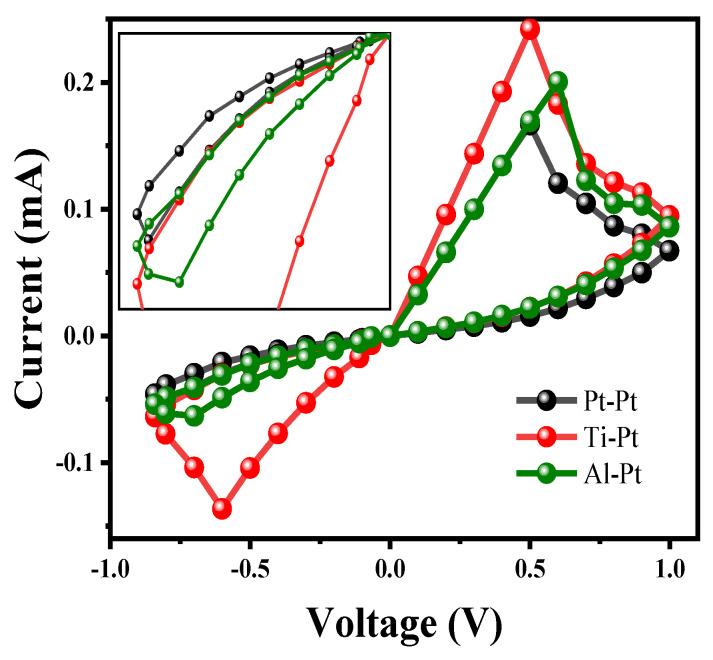
The I−V curves of three electrode combinations: Pt/HfO_2_/ZrO_2_/Pt, Ti/HfO_2_/ZrO_2_/Pt, and Al/HfO_2_/ZrO_2_/Pt.

**Figure 11 materials-17-01852-f011:**
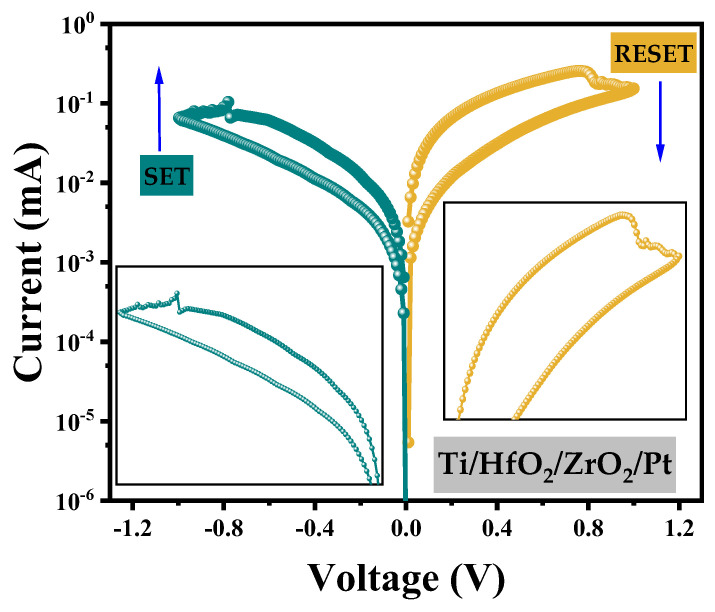
The I−V hysteresis loop of Ti/HfO_2_/ZrO_2_/Pt is applied at a bias of 0 V→1 V→0 V, where the sweep second rate is 100 mV/s.

**Table 1 materials-17-01852-t001:** Model parameters in COMSOL.

Parameters	Values	Description
E_a_ ^1^	1.5 [eV] [29]	Oxygen vacancy diffusion activation energy
k	1.380 × 10^−23^ [J/K]	Boltzmann constant
a ^1^	0.1 [nm] [30]	Ionic transition distance
f ^2^	1 × 10^13^ [ Hz] [30]	Escape frequency
q	1.602 × 10^−19^ [C]	The unit charge
A ^3^	1 × 10^24^ [cm^−3^ s^−1^] [31]	A constant associated with the frequency of vibration
b ^3^	0.75 [nm] [31]	Cell mesh size
E_c_ ^3^	0.8 [eV] [31]	Relaxation energy during recombination

^1^ The values of E_a_ and a refer to ref. [29]. ^2^ The values of f refer to ref. [30]. ^3^ The values of A, b, and E_c_ refer to ref. [31].

**Table 2 materials-17-01852-t002:** List of model size parameters in COMSOL [32].

Size Modules	Width (nm)	Thickness (nm)
Top Electrode (TE)	20	15
Conductive Filament 1 (CF1)	5	7.5
HfO_2_ switching layer (SL)	15	7.5
Conductive Filament 2 (CF2)	5	7.5
ZrO_2_ switching layer (SL)	15	7.5
Bottom Electrode (BE)	20	15

**Table 3 materials-17-01852-t003:** Lists the material properties of entities at each layer of the model in COMSOL. (Both HfO_2_ and ZrO_2_ used in the model are polycrystalline structures.)

Materials Parameters	Ti	Pt	HfO_2_	ZrO_2_
σ (S/m)	2.6 × 10^6^ [33]	8.9 × 10^6^ [34]	1 × 10^−6^ [31]	7.6 × 10^−6^ [35]
*k_th_* (W/m/K)	21.9 [33]	71.6 [34]	0.5 [36]	1.94 [37]
C_p_ (J/kg/K)	522	133	120 [31]	453 [37]
ρ (g/cm^3^)	4.506	21.450	9.680 [31]	6.100 [37]
ε_r_	1	1	25 [35]	20

Note. The parameter values of the cited references are referred to the corresponding reference, and the values of unlabeled parameters are provided by the COMSOL material library.

**Table 4 materials-17-01852-t004:** Comparison of our simulation findings with previously reported ZrO_2_ and HfO_2_ bilayer devices.

Device Structure	Deposition Technique	SET- Voltage (V)	RESET- Voltage (V)	Roff/Ron
TiN/ZrO_2_/HfO_2_/Pt [7]	ALD ^1^	−1.0 to −0.8	1.5 to 2.0	5
TaN/ZrO_2_/HO_2_/TiN [38]	Sputtering	1.0 to 0.8	−1. 0 to −0.8	10
TaN/HfO_2_/ZO_2_/Pt [18]	Sputtering	1.6 to 1.1	−1.4 to −0.9	14
Ti/HfO_2_/ZrO_2_/Pt This work	~	−0.7	0.6	10.46

^1^ ALD: atomic layer deposition.

## Data Availability

The raw data supporting the conclusions of this article will be made available by the authors on request.

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
