# Peer review of "Optimization of Bilayer Resistive Random Access Memory Based on Ti/HfO2/ZrO2/Pt"

_materials, 2024, doi:10.3390/ma17081852_

Round 1
Reviewer 1 Report
Comments and Suggestions for Authors
Overall, I find the manuscript well written and well organized with a lot of detail in the set-up for simulation. The topics of RRAM is promising future memory type and HfO2-type insulator is of particular interest to scientific community.
The simulations in the paper examine cylindrical symmetric Ti/HfO2/ZrO2/Pt layers at relevant thicknesses corresponding to other studies. Although the modelling approach follows that previously reported, the study examines unique HfO2/ZrO2 bilayer shown experimentally to be advantageous over HfO2 by itself. (ref 15) Simulations determined 2D mapping of oxygen vacancy, temperature, electric potential. Comparative simulations were done with different top electrodes, different widths/thicknesses of CF1 and CF2, to further understand mechanisms involved. From the simulation results “optimal” results were identified. I see relevant conclusions are made from the results of the study.
There is a closely related experimental study by Ismail et al., (Results in Physics (2020) 103274 using a TaN/HfO2/ZrO2/Pt which I would recommend including given the similarity of materials used.
To me, the weakest point of the paper is the lack of discussion how the simulation results in this study compare with others of either experimental or modelling where possible. I would recommend adding discussion on this. As an example, Ismail et al., has a nice table summarizing related studies although it doesn’t need to be that detailed.
What I would have included in the study would be comparison of HfO2/ZrO2 and only HfO2 to see if the simulation follows similar trends to that observed in ref 15. But I don’t think it is necessary to have this in this paper to be published. I was surprised to see on switch on/off transitions the changes all occur in the CF1 (HfO2). This would be good to note in the paper as well as any explanation why that may be. It is interesting to me how ZrO2 layer influences switching but doesn’t seem to change on transition.
Other more minor editorial points are as follows:
In abstract, “CF1” “ CF2” should be defined. Or better yet replace with HfO2 and ZrO2.
At end of section 1, “optimization” should stipulate based on simulation studies. Experimental findings may depend on more details for method of deposition etc.
In equation 2, G and R should be defined here, not later and equation 4, "a" should be defined here, not later.
In Tables 1-3, it seems references are cited here in brackets. Not sure I understand this. Are the reasons for choosing the parameters given found in these references. Adding a note under the tables to explain would be helpful.
Table 3 seems to have a typo in conductivity of HfO2. Please check this. Hope its not in the simulation that way. Also, equation 12, Table 3, P1 and P2 in text, and Figure 2 (x-axis) have inconsistencies with g/kg, m/cm. Please correct this.
Also in Table 3 where material parameters are given these values are different depending on phase of HfO2 and ZrO2 (monoclinic, cubic, tetragonal, amorphous, polycrystalline) which I suspect can vary the simulation results. So it be good to specify phase information in table if known. Or explain further in text about it.
Overall, good paper and I recommend publication after few changes noted above.
Author Response
The revision is in the attachment

Reviewer 2 Report
Comments and Suggestions for Authors
Manuscript materials-2919516 Round 1
Dr. Chunxia Wang manuscript is devoted to study the electrothermal coupling model of metal-oxide resistive random access memory (RRAM) is analyzed by using a 2-D axismetrical structure in COMSOL Multiphysics simulation software. The manuscript is theoretical in nature. Moderate editing of English language required. The manuscript is carelessly designed. The manuscript needs revision and cannot be published in this form.
1. The keywords are repeated in the title and abstract.
2. Line numbers are missing, making it much harder to see.
3. Page 1. Why right in the Abstract do I encounter some abbreviations CF1, CF2 that are not introduced or explained in any way?
4. Page 1. Last line, it says: Ti/Nb:STO. The abbreviation STO requires clarification. It should be mandatory to introduce any non-common abbreviations.
5. Pages 3, 4, 6. The formatting (presentation) of the equations raises a number of criticisms.
6. The review is poor. The authors should mention in the introduction other oxides including nanostructured Ta2O5 and Nb2O5 that are well suited for resistive memory 10.1142/S0218625X21500554.
7. Page 5. Section 2.2.2. Definition of Material Properties. Why is thermal conductivity are defined as a function of oxygen vacancy concentration nD? What are the heat conduction mechanisms underlying the model?
8. Page 6. Table 3. Is the specific conductivity of hafnium dioxide really 11 orders of magnitude higher than that of zirconium dioxide and only an order of magnitude worse than that of titanium and platinum? Such a striking difference of conductivities of zirconium and hafnium dioxides contradicts the information given on page 6 below the text. Are these the values that were used for the calculations? In that case, the validity of the results is not just questionable. On the contrary, the unreliability is not in doubt. I would add that in the cited source [35] (Basnet, P.; Pahinkar, D.G.; West, M.P.; Perini, C.J.; Graham, S.; Vogel, E.M. Substrate dependent resistive switching in amorphous-HfOx memristors: An experimental and computational investigation. J. Mater. Chem. C 2020, 8, 5092–5101, doi:10.1039/c9tc06736a.) I did not find specific values for the electrical conductivity of HfO2.
9. A general remark on the paper. A paper devoted to modeling without any experimental verification of the results is of very limited value. For example, the same cited paper [35] contains not only the results of calculations, but also a practical verification of their validity, demonstrating a rather high degree of coincidence between the calculated and experimental results. The reliability of the data of the proposed work is not proved in any way. It seems to me that experimental testing of at least several test structures performed in accordance with the recommendations of the authors would strengthen the practical significance of the results obtained.
10. The design of the figs is very poor.
11. The conclusion should be reworked. The conclusion should achieve the purpose of the paper as stated in the introduction and should consist of two parts. The first is the formulation of the general result achieving the goal with details and point by point important results that allowed to achieve the general result.
Comments on the Quality of English Language
Authors should improve the style, and most importantly work on the design of the text.
Author Response
The revision is in the attachment

Round 2
Reviewer 2 Report
Comments and Suggestions for Authors
The manuscript has gotten better and can be published.